# Roles of London Dispersive and Polar Components of Nano-Metal-Coated Activated Carbons for Improving Carbon Dioxide Uptake

**Seul-Yi Lee [1], Jong-Hoon Lee [1], Yeong-Hun Kim [1], Kyong-Yop Rhee [2],* and Soo-Jin Park [1],***

[1] Department of Chemistry, Inha University, Incheon 22212, Korea; leesy1019@inha.ac.kr (S.-Y.L.); boy834@naver.com (J.-H.L.); ihc2516@naver.com (Y.-H.K.)
[2] Department of Mechanical Engineering, College of Engineering, Kyung Hee University, Yongin 17104, Korea
* Correspondence: rheeky@khu.ac.kr (K.-Y.R.); sjpark@inha.ac.kr (S.-J.P.); Tel.: +82-31-201-2565 (K.-Y.R.); +82-32-876-7234 (S.-J.P.)

**Abstract:** Adsorption using carbonaceous materials has been considered as the prevailing technology for $CO_2$ capture because it offers advantages such as high adsorption capacity, durability, and economic benefits. Activated carbon (AC) has been widely used as an adsorbent for $CO_2$ capture. We investigated $CO_2$ adsorption behaviors of magnesium oxide-coated AC (MgO-AC) as a function of MgO content. The microstructure and textural properties of MgO-AC were characterized by X-ray diffraction and nitrogen adsorption–desorption isotherms at 77 K, respectively. The $CO_2$ adsorption behaviors of MgO-AC were evaluated at 298 K and 1 atm. Our experimental results revealed that the presence of MgO plays a key role in increasing the $CO_2$ uptake through the interaction between an acidic adsorbate ($e^+$) and an efficient basic adsorbent ($e^-$).

**Keywords:** London dispersive; polar component; magnesium oxide; activated carbon; carbon dioxide; adsorption

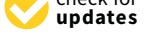



## 1. Introduction

Surface-free energy, which describes the physical phenomenon caused by intermolecular interaction at an interface, may be considered as the sum of two components: a dispersive component corresponding to London attraction and a nondispersive (or polar) component that includes all other types of interactions, such as Debye inductive force, Keesom orientational forces, hydrogen bonding, and Lewis acid–base interaction. In particular, knowledge of surface-free energy at a given temperature and surface enthalpy of a solid has resulted in advanced understanding in the field of adsorption, wettability, adhesion, etc. [1–3].

In the early 1960s, adsorption studies reported by Fowkes [4] and Graham [5] defined equilibrium spreading pressure ($\pi_e$) and surface-free energy of solids ($\gamma_S$) using simple adsorbates such as nitrogen, argon, and hydrocarbon. Park and Brendle [3] reported that an increase in the London dispersive component of surface-free energy ($\gamma_S^L$) is one of the main factors in the adsorption capacity, along with the specific surface area of the adsorbent including practically nonpolar adsorbents (for example, carbon materials). The specific (or polar) component of the surface-free energy ($\gamma_S^{SP}$) on the adsorbent is another important factor in increasing the strength of adsorption through acid–base interaction [6].

In particular, the molecular electric quadrupole moment of nonpolar molecules represents the first non-vanishing electric moment and thus dominates the interaction between the molecule and an external non-uniform electric field [7]. Since $CO_2$ has an electrical quadrupole moment of $-1.4 \times 10^{-39}$ C·m², which is much higher than those of other gases (e.g., $N_2$, $H_2$, etc.) [8,9], the $CO_2$ adsorption is largely governed by the inclusion of basic characteristics within the pore structure of the adsorbents, thereby inducing basicity and

an electric field in their cavities. The $CO_2$ molecule may interact with the electric fields of carbonaceous materials, leading to increased acid–base interaction [10].

Carbon dioxide ($CO_2$) is a critical heat-trapping greenhouse gas that causes global warming. Increasing the concentration of $CO_2$ in the atmosphere is assumed to directly result in global climate changes [11]. Considerable effort has been devoted to limiting the emissions of greenhouse gases into the environment. Technological advancements that address the global climate change through carbon capture and sequestration are as important as the innovations that lead to adoption of renewable energy [12,13].

Currently, commercially available $CO_2$ capture technologies are quite expensive and energy intensive. Improved technologies for $CO_2$ capture are required to achieve low energy penalties. Adsorption is considered to be a promising technology for capturing $CO_2$ from flue gases [14,15]. Adsorption methods to separate $CO_2$ from flue gases using physical adsorbents have been suggested. Solid adsorbents can be particularly effective for $CO_2$ separation from the synthesis gas. $CO_2$ adsorption on an adsorbent is of critical importance because there is a need to develop cost-efficient $CO_2$ capture technologies that can alleviate the consequences of climate change by improving gas separation processes of industrial interest [16,17]. The success of such an approach is strongly dependent on the use of suitable adsorbents with good $CO_2$ selectivity and adsorption capacity, resulting in a large specific surface area and viable $CO_2$-friendly sites on the adsorbent surfaces [18–21].

Among the porous materials available, activated carbon (AC) has been considered as an effective adsorbent for the removal of organic pollutants from wastewater and for gas purification owing to its large surface area and porosity [21,22]. In physical adsorption, the size and volume of the pores are important. Thus, microporous carbons are used for the sorption/separation of light gases, and porous carbons with large pore sizes are used for the removal of toxins or other large organic molecules. In contrast, to ensure specific interaction between an acidic adsorbate and basic adsorbents in the adsorption process, other features of AC, such as surface functional chemistry, should also be taken into consideration [23–26].

It is well-known that the forces of metals at the surface can be varied through modifications at ambient or elevated temperatures, which is associated with adsorbate bonding, geometry of the adsorption sites, etc. [27]. Various metal oxides have been identified as efficient post carbon capture sorbents [28–30]. Among them, magnesium oxide (MgO) has been recognized as a promising adsorbent candidate for $CO_2$ capture applications due to its suitable surface basicity, wide temperature range, and tunable physicochemical properties [31]. Moreover, hybrid adsorbents of MgO with metal compounds (alkali metal oxides, hydroxides, and carbonates/bicarbonates), and/or non-metals such as carbonaceous materials, silica, and metal–organic frameworks have a significant contribution in enhancing the $CO_2$ uptake performance [32,33].

In this work, MgO nanoparticles-coated AC are prepared by the metal-reduction method and a subsequent post-oxidation process to discuss the $CO_2$ adsorption mechanism. The $CO_2$ adsorption capacities are examined at 298 K and 1 atm, which are typically related to the flue gas of the post-combustion working conditions. The experimental details and results are discussed in the following sections.

## 2. Materials and Methods

Activated carbons (ACs, Tokyo Chemical Industry Co., Tokyo, Japan) were used as a neat material. Magnesium nitrate, sodium hydroxide, ethylene glycol, and formaldehyde were obtained from Sigma-Aldrich Co. (St. Louis, MO, USA) and used as received. Before loading magnesium oxide (MgO) onto the surfaces of the ACs, the ACs were purified by acid treatment using 5 M nitric acid solution to enhance the interfacial adhesion between magnesium and the AC surfaces.

For Mg coating, the ACs were suspended in an ethylene glycol (EG) solution under $N_2$ conditions. An EG solution containing Mg nitrate, with different loading amounts (1, 2, and 5 wt.%), was slowly added dropwise to the AC/EG solution with constant mechanical

stirring for 4 h under a $N_2$ atmosphere. A 1 M NaOH solution was added to adjust the pH of the solution, and 2 mL of formaldehyde solution as a reduction agent was then added dropwise into the solution. Then, the solution was heated to 120 °C for 2 h to completely reduce the Mg particles [34]. The MgO-coated ACs were prepared using a post-oxidation method on the Mg-coated AC samples. The oxidation temperature was fixed at 300 °C in an air stream for 10 min. The MgO-coated AC samples were labeled as a function of the Mg amount, as 1-MgO-ACs, 2-MgO-ACs, and 5-MgO-ACs.

The MgO content on the AC surfaces was examined using X-ray photoelectron spectroscopy (XPS, ESCA LAB MK-II, VG Scientific Co., East Grinstead, UK) with monochromatic Al Kα radiation (hυ = 1486.6 eV), operated under the constant analysis mode. The survey spectra were measured in the range of 50–1350 eV pass energy. A non-linear least squares curve-fitting program (Peak-Fit version 4, Systat Software Inc., San Jose, CA, USA) with a Gaussian-Lorentzian mix function and Shirley background subtraction was used to de-convolve the XPS subpeaks [35]. An elemental analyzer (EA, EA1112, Thermo Scientific Co., Waltham, MA, USA) was used to investigate chemical elemental composition of the prepared samples. X-ray diffraction (XRD, Model D/MAX-II, Rigaku Co., Tokyo, Japan), which was performed using Cu Kα radiation, showed changes in the crystalline phase of the prepared adsorbents and lattice distortions. An indexing identification of the peak and a peak-fitting analysis (Xprocess Program, USA) was performed to identify the diffraction of nanoparticles for the determination of the sizes. The $N_2$ adsorption isotherms were measured at 77 K using a gas adsorption analyzer (BEL-SORP, Bel Co., Tokyo, Japan). The samples were outgassed at 383 K for 24 h to obtain a residual pressure less than $10^{-3}$ torr. The specific surface areas and micropore volumes of the samples were determined from the BET equation and t-plot method, respectively. The mesoporous volumes of the samples were determined using the BJH method [36]. The amount of $N_2$ adsorbed at relative pressures ($P/P_0$ = 0.98) was used to determine the total pore volume, which corresponds to the sum of the micropore and mesopore volumes.

The $CO_2$ adsorption performance was evaluated at 298 K using BEL-SORP (BEL Co., Tokyo, Japan). The samples were outgassed at 393 K for 24 h to obtain a residual pressure less than $10^{-3}$ torr. After degassing under vacuum, the $CO_2$ adsorption of the sorbents was evaluated at 298 K and 1 atm.

## 3. Results and Discussion

### 3.1. Morphological Properties

Figure 1 shows the SEM images of the MgO nanoparticle-coated AC surfaces with different MgO loading amounts. With the increasing MgO amount, the roughness of the sample surfaces appeared to increase because the MgO particles were introduced onto the AC surfaces. This means that the AC surfaces could be blocked by the Mg particles and the subsequent post-oxidation process.

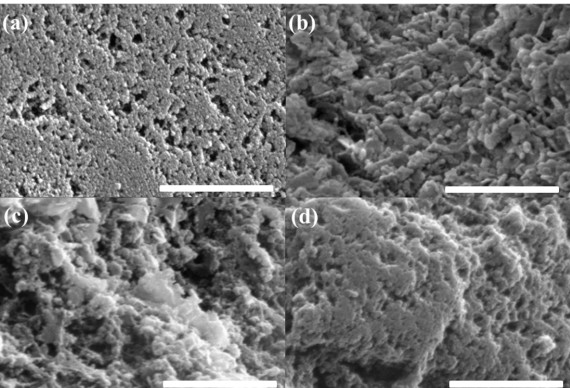

**Figure 1.** SEM images of (**a**) bare AC, (**b**) 1-MgO-AC, (**c**) 2-MgO-AC, and (**d**) 5-MgO-AC (scale bar: 500 nm).

### 3.2. Surface Properties

XPS was used to study the chemical composition of MgO-coated AC surfaces with different MgO loading amounts. Table 1 lists the atomic percentage of Mg that increased gradually with the increasing MgO loading amounts on the AC surfaces. Figure 2 presents the high-resolution O1s and Mg2p subpeaks of the prepared samples. These data revealed that three O1s subpeaks can be easily distinguished in the XPS spectra of all the samples. The peaks of MgO, Mg(OH)$_2$, and MgCO$_3$ have binding energies in the ranges of 530.0–531.0 eV, 530.0–533.2 eV, and 533.2–533.5 eV, respectively [37,38]. We found that MgO is predominantly formed during post-oxidation at 300 °C due to the reaction with air. The four subpeaks in the ranges of 49.5–49.8 eV, 50.5–50.8 eV, 51.0–51.5 eV, and 51.9–52.5 eV corresponded to metallic Mg, MgO, Mg(OH)$_2$, and MgCO$_3$, respectively. These peak assignments agree with the values reported in the literature [39,40]. The intensities of the Mg2p peaks increased with the increasing MgO amounts, which indicates that the increased intensity of the Mg2p peak can be mainly attributed to the MgO formation, as shown in Figure 2f. This means that the MgO-coated AC samples possess Mg$^{2+}$ ions, which induce a strong affinity toward CO$_2$ at low pressures [41–43].

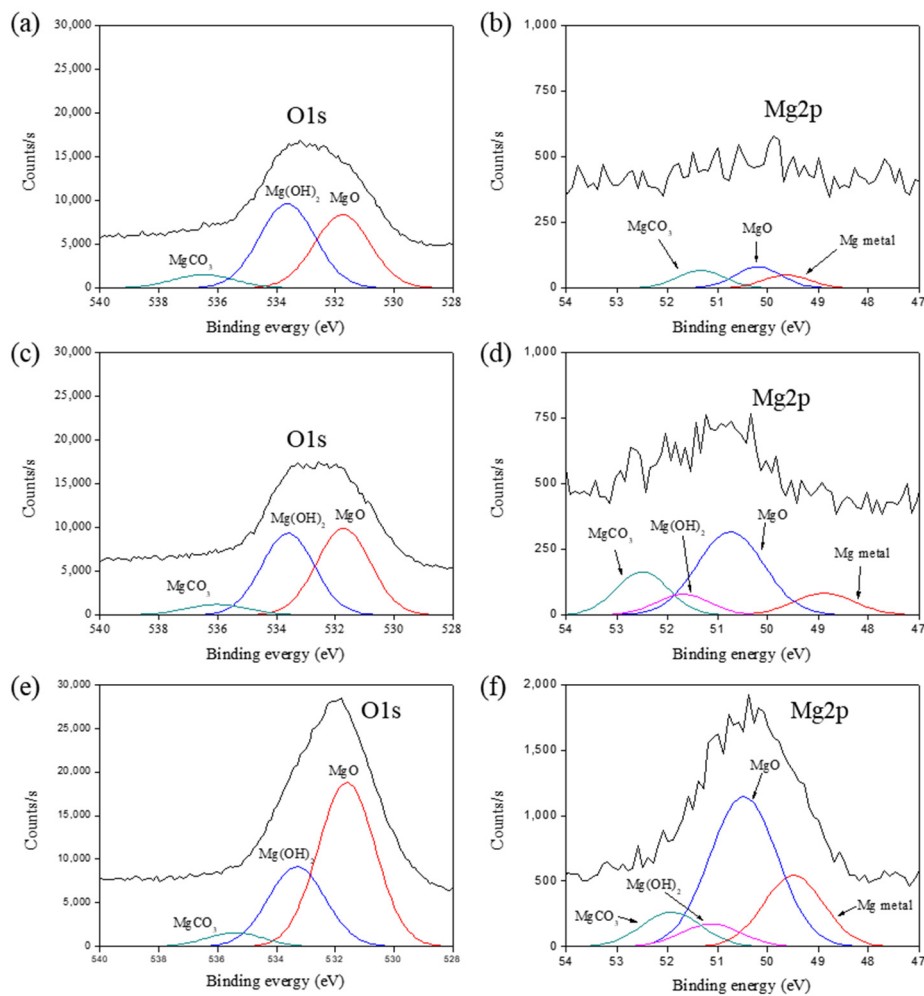

**Figure 2.** High-resolution peaks of (**a**,**b**) 1-MgO-AC, (**c**,**d**) 2-MgO-AC, and (**e**,**f**) 5-MgO-AC.

**Table 1.** Chemical compositions and textural properties of the MgO-coated AC samples.

| Specimens | Atomic Percentage (at.%) | | | Textural Properties | | | |
|---|---|---|---|---|---|---|---|
| | C | O | Mg | $S_{BET}$[1] | $V_{total}$[2] | $V_{micro}$[3] | $V_{meso}$[4] |
| AC | 85.4 | 12.2 | - | 1608 | 1.57 | 0.20 | 1.37 |
| 1-MgO-AC | 81.2 | 15.8 | 0.87 | 1203 | 1.00 | 0.19 | 0.81 |
| 2-MgO-AC | 78.9 | 15.7 | 2.55 | 1123 | 0.96 | 0.16 | 0.75 |
| 5-MgO-AC | 63.8 | 22.9 | 6.72 | 1096 | 0.74 | 0.13 | 0.61 |

$S_{BET}$[1]: specific surface area ($m^2/g$), $V_{total}$[2]: total pore volume ($cm^3/g$), $V_{micro}$[3]: micropore volume ($cm^3/g$), $V_{meso}$[4]: mesopore volume ($cm^3/g$)

### 3.3. Structural Properties

Figure 3 shows the XRD patterns of the MgO-coated ACs prepared with different MgO amounts. All the samples showed prominent peaks at 2 theta = 26°, which is a typical value for crystalline graphite of ACs. With the increasing MgO content, the intensity of the carbon peaks of the ACs decreased slightly. The diffraction peaks of the MgO-coated AC samples at 2 theta = 43° and 62° were assigned to (200) and (220) of MgO, respectively [44]. Furthermore, the intensity of the MgO diffraction peaks was enhanced gradually with the increasing MgO fractions.

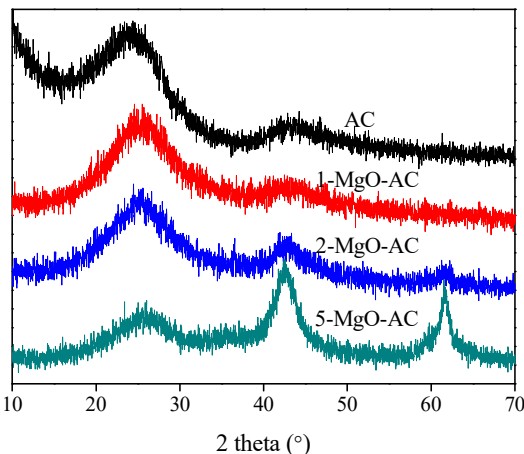

**Figure 3.** XRD patterns of bare AC and MgO-coated AC samples.

The mean particle sizes of the MgO particles on the AC surfaces were calculated from the XRD patterns using the Scherrer formula:

$$L_c = \frac{K\lambda}{\beta_{1/2}\cos\theta} \tag{1}$$

where $L_c$ is the mean particle size (nm), $K$ the Scherrer constant (=0.9), $\lambda$ the X-ray wavelength (Cu K$\alpha$ = 0.154 nm), $\theta$ the angle at peak maximum, and $\beta_{1/2}$ the full width (radians) of the peak at half height [45].

According to the Scherrer equation, the mean crystallite sizes of MgO in 1-MgO-ACs, 2-MgO-ACs, and 5-MgO-ACs are approximately 8.1, 8.5, and 9.0 nm, respectively. The crystallite sizes of MgO increased slightly with the increasing MgO amounts, which affected the specific surface and pore volumes of the MgO-coated AC samples.

### 3.4. Textural Properties

To explore the textural properties, $N_2$/77 K adsorption–desorption isotherms were evaluated, as presented in Figure 4. Bare AC showed a typical type I isotherm with significant adsorption below a relative pressure $P/P_0 = 0.01$, according to IUPAC classification, indicating a microporous structure [46,47]. However, the adsorption isotherms of the MgO-

coated AC samples displayed type IV behavior with a low $N_2$ uptake at lower pressures, indicating capillary condensation in the mesopores (>2 nm). Moreover, the $N_2$ adsorption capacities decreased with increasing the MgO amounts. This can be attributed to the pore filling or blocking by the MgO coating onto the AC surfaces.

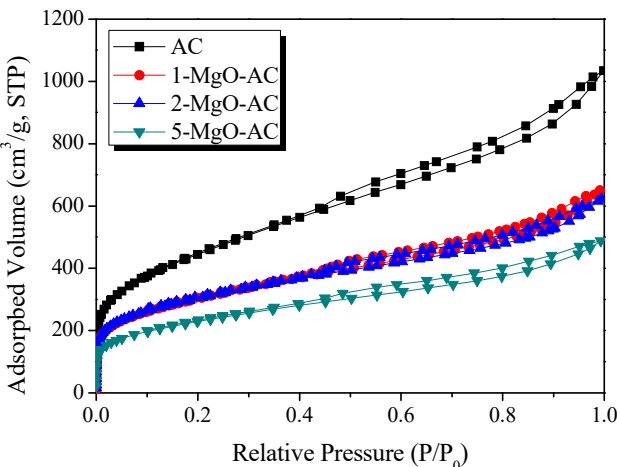

**Figure 4.** $N_2$/77 K adsorption–desorption isotherms of AC and MgO coated ACs with different MgO loading amounts.

As listed in Table 1, the textural properties indicated that all the samples exhibited mesoporous structures. The bare AC showed the highest specific surface area, micropore volume, and mesopore volume of 1608 $m^2/g$, 0.2 $cm^3/g$, and 1.37 $cm^3/g$, respectively. With the increasing MgO loading amounts, the specific surface area and mesopore volume decreased gradually while the micropore volume did not change significantly. From the results, it is clear that the MgO coating significantly decreases the mesopores on the AC surfaces, resulting in a decrease in the specific surface area.

### 3.5. $CO_2$ Uptake Behaviors

The $CO_2$ adsorption isotherms of bare AC and MgO-coated AC samples were measured at 298 K and 1 atm through pressure swing analysis, as shown in Figure 5. The $CO_2$ adsorption capacities of AC, 1-MgO-AC, 2-MgO-AC, and 5-MgO-AC were 74.4, 83.3, 108.8, and 105.1 mg/g, respectively. The MgO-coated AC samples showed a higher adsorption capacity than bare AC. This clearly indicates that MgO plays a key role in improving the $CO_2$ adsorption capacity. The adsorption capacities were found to increase with the increasing MgO loading amounts up to 2-MgO-AC and showed no significant change in 5-MgO-AC. The optimum effect of MgO coating can be assessed from the bare AC and 2-MgO-AC samples; the latter was approximately 32% higher (2.55 at.% MgO fraction obtained from XPS data) than the former (without treatment). This can be attributed to an enhanced interaction between the $CO_2$ molecule and MgO through the acid ($\delta^+$)-base ($\delta^-$) interaction, which is referred to as the specific (polar) component ($\gamma_S^{SP}$) of the surface-free energy on the adsorbent, resulting in improved $CO_2$ adsorption uptake.

We noted that the $CO_2$ adsorption capacity of 5-MgO-AC showed no improvement even though 5-MgO-AC had a considerable MgO loading amount (6.72 at.% of Mg obtained from the XPS data). The experimental data indicate that the higher loading amounts of MgO did not improve the adsorption capacity of the MgO-coated AC, probably owing to the decreased specific surface area and pore volume because the excessive MgO coating resulted in the filling and/or blocking of the pores on the AC surfaces. This is not favorable for enhancing the $CO_2$ adsorption capacity because of the inadequacy of the London dispersive component of the surface-free energy ($\gamma_S^L$). Thus, it appears that the lack or minimal presence of the specific surface area and pore structures may take away the expected benefit of the interaction between the adsorbent and adsorbate.

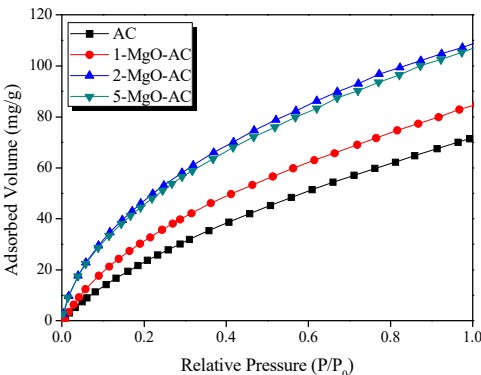

**Figure 5.** $CO_2$ adsorption capacity of the samples studied at 298 K and 1 atm.

Therefore, the net result is an increase in the $CO_2$ adsorption capacity of the acidic adsorbate and efficient basic adsorbent, resulting in increased the polar-polar interaction, in the presence of the specific surface area and pore volume of ACs. The schematic illustrations of the suggested $CO_2$ adsorption mechanism on the bare AC and MgO-coated AC are shown in Figure 6.

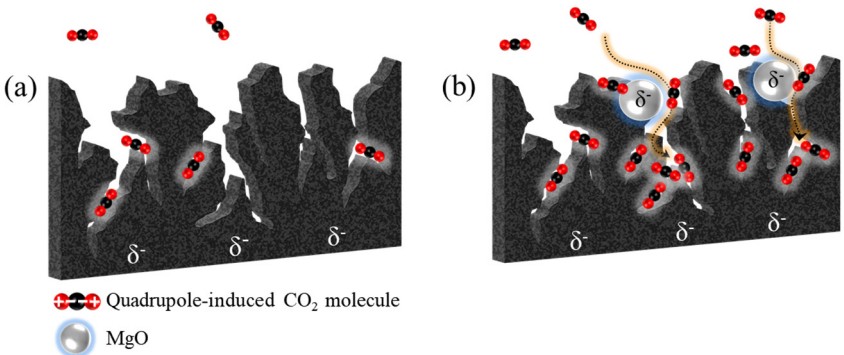

**Figure 6.** Schematic illustrations of the suggested $CO_2$ adsorption mechanism on (**a**) bare AC and (**b**) MgO-coated AC.

## 4. Conclusions

In this work, we studied the $CO_2$ adsorption behaviors of MgO-coated AC as a function of MgO content. The experimental results showed that the $CO_2$ adsorption capacity of the prepared samples increased with the increasing MgO loading amounts up to 2-MgO-AC owing to an improvement of acid–base interaction (refer to the specific (polar) component of the surface-free energy ($\gamma_S^{SP}$)). Equally important, the data suggested that the specific surface areas and pore volumes are the key factors in improving the $CO_2$ adsorption capacity, resulting from the London dispersive component of surface-free energy ($\gamma_S^L$). Thus, our results on the underlying physics of adsorption mechanism should help in designing adsorbents for gases more efficiently, particularly those with quadrupole moments.

**Author Contributions:** Conceptualization and methodology, S.-Y.L., J.-H.L. and Y.-H.K.; Writing—original draft preparation, S.-Y.L., J.-H.L. and Y.-H.K.; Writing—review and editing, K.-Y.R. and S.-J.P.; Supervision, K.-Y.R. and S.-J.P. All authors have read and agreed to the published version of the manuscript.

**Funding:** This work was supported by Nano-Convergence Foundation(www.nanotech2020.org) funded by the Ministry of Science and ICT(MSIT, Korea) & the Ministry of Trade, Industry and Energy (MOTIE, Korea) [Development of high-efficiency activated carbon filter for removing indoor harmful elements (VOCs, radon, bacteria, etc.)], supported by Korea Evaluation institute of Industrial

Technology (KEIT) through the Carbon Cluster Construction project [10083586, Development of petroleum based graphite fibers with ultra-high thermal conductivity] funded by the Ministry of Trade, Industry & Energy (MOTIE, Korea), and supported by Korea Electric Power Corporation [R21XO01-5].

**Institutional Review Board Statement:** Not applicable.

**Informed Consent Statement:** Not applicable.

**Data Availability Statement:** Data is contained within the article.

**Conflicts of Interest:** The authors declare no conflict of interest.

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
