# Peer review of "Roles of London Dispersive and Polar Components of Nano-Metal-Coated Activated Carbons for Improving Carbon Dioxide Uptake"

_coatings, doi:10.3390/coatings11060691_

Round 1

Reviewer 1 Report

The manuscript presented by Seul-Yi Lee and co-authors about Roles of London Dispersive and Polar Components of Nano-metal-coated Activated Carbons for Improving Carbon Dioxide Uptake, the manuscript looks quite simple, only four samples were studied, and the analysis looks also not so well, for example, XPS analysis results are not so reasonable, how to make deconvolution based on this peak in Figure 2b, the peaks looks like noise peaks, and it is easy for readers to doubt the results. The authors should show novelty for readers in the manuscript. Some sentences were written not so well, the English should be improved, and also the mistakes in the content and reference should be corrected.

Author Response

[ Reviewer#1]

The manuscript presented by Seul-Yi Lee and co-authors about Roles of London Dispersive and Polar Components of Nano-metal-coated Activated Carbons for Improving Carbon Dioxide Uptake, the manuscript looks quite simple, only four samples were studied, and the analysis looks also not so well, for example, XPS analysis results are not so reasonable, how to make deconvolution based on this peak in Figure 2b, the peaks looks like noise peaks, and it is easy for readers to doubt the results. The authors should show novelty for readers in the manuscript. Some sentences were written not so well, the English should be improved, and also the mistakes in the content and reference should be corrected. 

  • Thank you for your review. The present study aims to investigate the roles of London dispersive and Polar components of MgO/AC on the CO2 adsorption behaviors. From the results of four MgO/AC samples obtained by different MgO loading amounts, we found the loading amounts of MgO have affected on London dispersive component as well as Polar (specific) component of the AC surfaces, which play an important role in the interaction between CO2 (adsorbate) and MgO/AC (adsorbent). Our novelty is to see the CO2 adsorption of nanometallic particles-loaded AC from a new angle of the surface free energy. We believe this should help design the efficient adsorbent for gases that, particularly, have quadrupole moment.
  • Yes, we agree that the intensity of 1-MgO-AC sample is very low due to a little Mg amount (0.87 at.%), but it was found that the intensities of Mg peaks are increased as useful as to show the chemical compositions of Mg (or MgO) surfaces in 2-MgO-AC and 5-MgO-AC. From the XPS results, we found the subpeaks of MgO were getting much bigger, which is a key role in improving the CO2 adsorption capacity, than other components such as Mg(OH)2, MgCO3.
  • The revised manuscript is now professionally edited by American native speaker. Please see the Certificate of Proof-reading provided by the Editing Agency (please see the next page).

Reviewer 2 Report

The manuscript titled "Roles of London Dispersive and Polar Components of Nano- metal-coated Activated Carbons for Improving Carbon Dioxide Uptake" done by Seul-Yi Lee, Jong-Hoon Lee, Yeong-Hun Kim, Kyong Yop Rhee, and Soo-Jin Park treats about an investigation of CO2 adsorption on Active Carbon coated by MgO. It is appropriately written and performed measurements well document the results. It deserves publication; however, I have some remarks on the manuscript content. Firstly to the procedure of ac coating, line 98 and 99, the EG solution had contained the Mg nitrate, or instead, the Mg nitrate was slowly added by dropwise method to the solution of EG. For me, it is not clear; can it be clarified? Table 1 is suddenly appearing in the text body. My question is: haw was evaluated the atomic concentration in Table. The authors' nothing to say about it. In my opinion, the calculation of MgO particle sizes is burdened with a significant error. In Figure 3 it is seen that the peak of AC pattern at 2 theta close to 43 deg is covering with the peak from MgO particle. This results in peak brooding and artificially decreases the size of particles. I understand these calculations are not crucial for the manuscript, but it will be better when the authors could insert some comment about it. My last note concerns English, it is mostly correct, but at the end of the manuscript, the authors start using long sentences (lines 232 to 235 and 237 to 239), which makes the findings vague. A set of words: comparably considerable, in line 231 is also hard to understand. 

Author Response

The manuscript titled "Roles of London Dispersive and Polar Components of Nano- metal-coated Activated Carbons for Improving Carbon Dioxide Uptake" done by Seul-Yi Lee, Jong-Hoon Lee, Yeong-Hun Kim, Kyong Yop Rhee, and Soo-Jin Park treats about an investigation of CO2 adsorption on Active Carbon coated by MgO. It is appropriately written and performed measurements well document the results. It deserves publication; however, I have some remarks on the manuscript content.

è Thank you for your kind review.

  1. Firstly, to the procedure of ac coating, line 98 and 99, the EG solution had contained the Mg nitrate, or instead, the Mg nitrate was slowly added by dropwise method to the solution of EG. For me, it is not clear; can it be clarified?

è We changed the sentence for readers to understand clearly. Please see the P3/L98-100.

< Revised Sentence >

An EG solution containing Mg nitrate, with different loading amounts (1, 2, and 5 wt.%), was slowly added dropwise to the AC/EG solution with constant mechanical stirring for 4 h under a N2 atmosphere

  1. Table 1 is suddenly appearing in the text body. My question is: haw was evaluated the atomic concentration in Table. The authors' nothing to say about it. In my opinion, the calculation of MgO particle sizes is burdened with a significant error. In Figure 3 it is seen that the peak of AC pattern at 2 theta close to 43 deg is covering with the peak from MgO particle. This results in peak brooding and artificially decreases the size of particles. I understand these calculations are not crucial for the manuscript, but it will be better when the authors could insert some comment about it.

è We added the measurement method for elemental analysis in the Experimental Section. Please see P3/L112-114.

< Added sentence >

An elemental analyzer (EA, EA1112, Thermo Scientific Co.) was used to investigate chemical elemental composition of the prepared samples.

è We used the software (Xprocess Program) to identify the peak of MgO in the brood. We added the sentence in the Experimental Section according to the Reviewer’s comment. Please see P3/L116-118.

< Added sentence >

An indexing identification of the peak and a peak-fitting analysis (Xprocess Program) was performed to identify the diffraction of nanoparticles for the determination of the sizes.

  1. My last note concerns English, it is mostly correct, but at the end of the manuscript, the authors start using long sentences (lines 232 to 235 and 237 to 239), which makes the findings vague. A set of words: comparably considerable, in line 231 is also hard to understand.

è According to the Reviewers’ comment, the revised manuscript has been professionally edited by American native speaker. Please see the Certificate of Proof-reading provided by the Editing Agency as enclosed in this copy.

Reviewer 3 Report

The CO2 adsorption behaviour of MgO-AC were evaluated. The experimental resultsreveal that the presence of MgO play a key role in increasing the CO2 uptake by the interaction between acidic adsorbate and basic adsorbent. The manuscript contains useful information and it can be published after minor correction.

  1. Please incorporporate little bit more basic literature dealing with the nature of CO2 activation (for example: Surf. Sci. 207, (1988) 36-54);  Surf. Sci. Rep.  71, (2016) 595-671).
  2. What was the reference in XPS measurements.
  3. Please give reference for metallic Mg in XPS.

Author Response

The CO2 adsorption behavior of MgO-AC were evaluated. The experimental results reveal that the presence of MgO play a key role in increasing the CO2 uptake by the interaction between acidic adsorbate and basic adsorbent. The manuscript contains useful information and it can be published after minor correction.

  1. Please incorporporate little bit more basic literature dealing with the nature of CO2 activation (for example: Surf. Sci. 207, (1988) 36-54); Surf. Sci. Rep. 71, (2016) 595-671).
  • Thank you for recommending the literatures. We added new references in the revised manuscript according to the Reviewer’s comment. Please see the Page 2/L76-78 and L85.

< Added sentence and references [27], [33] >

It is well-known that the forces of metals at the surface can be varied through modifications at ambient or elevated temperatures, which is associated with adsorbate bonding, geometry of the adsorption sites, etc. [27].

[27] Bohnen, K.P.; Ho, K.M. Structure and dynamics at metal surfaces. Surface Science Reports 1993, 19, 99-120

[33] Taifan, W.; Boily, J.-F.; Baltrusaitis, J. Surface chemistry of carbon dioxide revisited. Surface Science Reports 2016, 71, 595-671

  1. What was the reference in XPS measurements.

è We added the reference on the XPS measurements and deconvolution technique in the Experimental Section in the revise manuscript. Please see the Page 3/L 110-112.

< Added sentence and references [35] >

A non-linear least squares curve-fitting program (Peak-Fit version 4) with a Gaussian-Lorentzian mix function and Shirley background subtraction was used to de-convolve the XPS subpeaks [35]. 

[35] Koenig, M.F.; Grant, J.T. Deconvolution in X-ray photoelectron spectroscopy. Journal of Electron Spectroscopy and Related Phenomena 1984, 33, 9-22

  1. Please give reference for metallic Mg in XPS.

è We added two references in the revised manuscript. Please see the Page 4/L148-151.

< Added sentence and references [39,40] >

The four subpeaks in the range of 49.5-49.8 eV, 50.5-50.8 eV, 51.0-51.5 eV, and 51.9-52.5 eV were corresponded to metallic Mg, MgO, Mg(OH)2, and MgCO3, respectively. These assignments are in an agreement with the values reported in literature [39,40].

[39] Ardizzone, S.; Bianchi, C.L.; Fadoni, M.; Vercelli, B. Magnesium salts and oxide: an XPS overview. Applied Surface Science 1997, 119, 253-259

[40] Müller, M.; Matthes, F.; Schneider, C.M. Photoemission study of the Fe(001)∕MgO interface for varying oxidation conditions of magnesium oxide. 2007, 101, 09G519

Round 2

Reviewer 1 Report

This manuscript is better now after revision, the authors had already considered some comments and made the modification. There are still some small things which could be modified to better, e.g., the original scale bars in the SEM images were not clear, new clear scale should be made. The manuscript would be better if the authors could make more scans for the XPS analysis, then the peaks could be more readable and the data could be more precise.

Author Response

We would like to thank the reviewers for their in-depth reviews and excellent comments/suggestions regarding our manuscript. In the following, we provide response to each of the reviewers’ comments in the exact order as received.

Reviewer 2 Report

According to the introduced changes by the authors, I find that the manuscript  is ready for publication.

Author Response

According to the introduced changes by the authors, I find that the manuscript is ready for publication.

 Thank you very much for your time.